# Effects of Yb Addition on the Microstructure and Mechanical Properties of As-Cast ADC12 Alloy

**Junjie Xiong** [1,2], **Hong Yan** [1,2,*] , **Songgen Zhong** [1,2] and **Minzhu Bi** [3]

[1] School of Mechanical Electrical Engineering, Nanchang University, Nanchang 330031, China; 15070995216@163.com (J.X.); zsg9555@163.com (S.Z.)
[2] Key Laboratory of Light Alloy Preparation & Processing, Nanchang University, Nanchang 330031, China
[3] Jingdezhen Mingxing Aviation Forging Co., Ltd., Jingdezhen 333403, China; bmzhd@126.com
[*] Correspondence: hyan@ncu.edu.cn; Tel.: +86-791-8396-9633; Fax: +86-791-8396-9622

**Abstract:** The effects of addition of different amounts of rare earth ytterbium (Yb) on the microstructure and mechanical properties of the casting ADC12 alloy were investigated by mechanical properties testing and microstructure observation. The results indicate that Yb modification had a big influence on the microstructure and properties of the as-cast alloy. The optimum level of Yb content was 0.8 wt %. The coarse dendritic primary α-Al phases were fully refined, leading to the decreasing of the secondary dendrite arm spacing. The morphology of eutectic silicon phases changed from acicular into short rod-like and even granular. There was a structural transformation of β-Fe phases from massive to small rod-shaped morphology. Additionally, the tensile strength, elongation, and microhardness were 267.9 MPa, 4.2%, and 107.3 HV, respectively, increases of 55.4, 121.1, and 41.4%, respectively, compared with the matrix alloy. Fractographic examinations reveal that mainly ductile fracture for Yb addition of 0.8 wt %. The fracture appearances matched the tendency of the tensile properties. Furthermore, the addition of Yb can generate a rare earth phase consisting of the three elements of Al, Si, Yb, with some small iron-rich phases attached around the rare earth phase.

**Keywords:** rare earth Yb; ADC12; modifying; microstructure and properties; fracture

## 1. Introduction

Lightweight Al-Si alloys are extensively used in the automotive industry to reduce vehicle weight. In order to enhance the overall quality and performance of these vehicles, higher requirements are placed on alloy properties to improve overall performance. Good performance of alloy materials requires fine microstructures, and modification is widely applied as a strategy to refine the microstructure of Al-Si alloys [1–4].

Rare earth elements are often used to modify Al-Si alloys, and only small amounts are needed to provide significant improvement of alloy properties. [5,6]. Rare earth materials are abundant and there are many different kinds. Thus, there is strong interest in testing these materials for the modification of Al-Si alloys to obtain properties of interest. Hu et al. found that Sm addition effectively refined the microscopic structure of as-cast Al-Si-Cu alloy effectively. The morphology features of α-Al dendrite, eutectic silicon, and the iron-rich phases were all improved. In addition, AlSiSm and AlSiCuSm intermetallics were identified [7]. Song et al. observed a decrease of 50% in the mean diameter of Al-Si-Mg alloy with Pr/Ce addition of 0.6 wt %, resulting in an alloy with good comprehensive performance [8]. R. Ahmad et al. investigated the transformation of the Al-Si-Cu-Mg alloy with different Ce concentrations. The results demonstrated the alloy containing 0.1 wt % Ce exhibited a low

temperature and fast solidification time of the α-Al phase, significantly decreasing the temperature and time of the matrix alloy [9].

The ADC12 alloy is extensively used in the aerospace and automobile industries due to its excellent properties, such as high strength, high fluidity, fine corrosion resistance, and good castability [10–13]. Eutectic silicon phases and iron-rich phases are main secondary reinforcing phases in the ADC12 alloy, and these phases can be refined by the addition of rare earths. However, the transformative effect of rare earth elements is related to do atomic radius. The optimal ratio of the atomic radius of the rare earth element and the atomic radius of Si is about 1.646 [14]. For Yb, the atomic radius ratio between Yb and Si is 1.66, close to the optimal ratio, suggesting Yb will exhibit good modification of Si phases. Thus, modification of ADC12 with Yb addition to improve alloy performance is a reasonable strategy. This study investigated the effects of different Yb additions on the microstructure and properties of ADC12 alloy, followed by discussion of the modification mechanism.

## 2. Experimental

ADC12 alloy was used as the matrix. Al-10Yb master alloy was used as the additive phase. The graphite crucible (60 mm diameter and 100 mm height) was preheated to 200 °C, and then the ADC12 alloy was melted in the crucible at 750 °C by using an electrical resistance furnace. A sample of 180 g aluminum alloy was used for melting each time. The prepared Al-10Yb wt % master alloy was added into the melt to prepare alloy with different contents of Yb (0.0, 0.4, 0.8, and 1.2 wt %) after the matrix alloy was completely melted. The melt was isothermally held at 720 °C for about 30 minutes to ensure the complete dissolving of the Yb. Then, the surface of the melt was skimmed and poured into the permanent mold which was preheated to 300 °C. The aluminum melt pool was protected by argon gas throughout the whole experiment. The quality fraction of rare earth in the ADC12 alloy was determined by using ICP-AES (inductively coupled plasma atomic emission spectrometry, Beijing Huake ETS Analysis Instrument Co., Ltd., Beijing, China) testing, as shown in Table 1.

The cast samples were processed into standard tensile test bars according to the standard of GB/T228-2002, and tensile tests were carried out using a UTM5105 testing machine (Zhuhai SUST Electrical Equipment Co., Ltd., Zhuhai, China) at a crosshead speed of 1 mm/min. The shape and dimensions of the tensile specimens are schematically shown in Figure 1. For statistical confidence, five test specimens of each sample were tested, and the average value was determined. Specimens for microstructure examination were prepared from the as-cast samples and etched using a solution of 0.5 vol % hydrofluoric acid. Optical microscopy (OM, Eclipse MA200, Nikon Metrology, Inc., Brighton, UK) and a scanning electron microscope (SEM, VEGA3, TESCAN CHINA, Ltd., Shanghai, China) equipped with an energy diffraction spectrum (EDS, Oxford Instrument Co., Ltd., Oxford, UK) were used to observe the microstructure of the prepared Yb-containing alloys. The tensile fracture surfaces were analyzed by SEM (FEI Trading (Shanghai) Co., Ltd., Shanghai, China). The microhardness of each sample was measured 10 times with a HV-1000A Vickers hardness tester (Laizhou Huayin Testing Instrument Co., Ltd., Laizhou, China) with a load of 300 gf and a holding time of 10 s. Two parameters (mean area and aspect ratio) of the secondary phase were calculated with Image-Pro Plus 6.0 analysis software (Guangzhou Micro-shot Technology Co., Ltd., Guangzhou, China). Approximately 30 different areas of each microstructure were measured to minimize errors. The two parameters were defined according to the following equations:

$$\text{Mean area} = \frac{1}{m}\sum_{j=1}^{m}\left(\frac{1}{n}\sum_{i=1}^{n}A_i\right)_j \qquad \text{Aspect ratio} = \frac{1}{m}\sum_{j=1}^{m}\left[\frac{1}{n}\sum_{i=1}^{n}\left(\frac{L_l}{L_s}\right)\right]_j \tag{1}$$

where $A_i$ is the area of a single secondary phase, $L_l/L_s$ is the ratio of the longest to the shortest dimensions of a single secondary phase, $n$ is the number of particles of a single field, and $m$ is the number of fields.

**Table 1.** Chemical composition of testing samples (mass fraction/wt %).

| Number | Materials | Si | Cu | Mg | Fe | Zn | Yb | Al |
|--------|-----------|------|------|------|------|------|------|------|
| 1 | ADC12 | 10.76 | 3.02 | 0.31 | 0.62 | 0.54 | 0 | Bal. |
| 2 | 0.4 wt %Yb/ADC12 | 10.58 | 3.13 | 0.34 | 0.59 | 0.62 | 0.43 | Bal. |
| 3 | 0.8 wt %Yb/ADC12 | 10.62 | 3.08 | 0.29 | 0.63 | 0.61 | 0.84 | Bal. |
| 4 | 1.2 wt %Yb/ADC12 | 10.67 | 2.97 | 0.36 | 0.57 | 0.58 | 1.18 | Bal. |

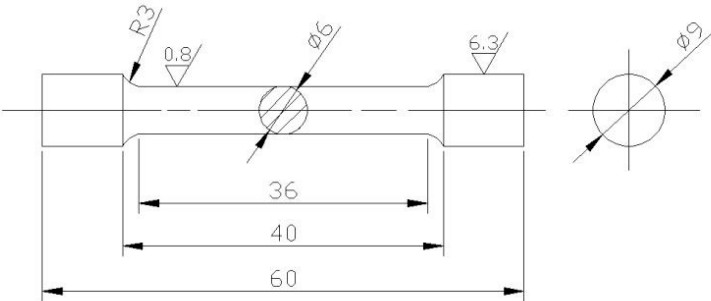

**Figure 1.** Schematic diagram of the standard tensile specimen (unit: mm).

## 3. Results and Discussion

### 3.1. Effects of Yb Addition on α-Al Phases

The optical microstructures of as-cast ADC12 alloys with different Yb contents were determined and are shown in Figure 2. In Figure 2a, the coarse dendritic microstructure can be clearly observed in the unmodified ADC12 alloy. The α-Al grains in the modified ADC12 alloys are finer and have small secondary dendrite arm spacing (SDAS), compared with that of the unmodified ADC12 alloy, as shown in Figure 2b–d. It can be clearly seen that the refinement of α-Al with 0.8 wt % Yb is the most effective, as shown in Figure 2c.

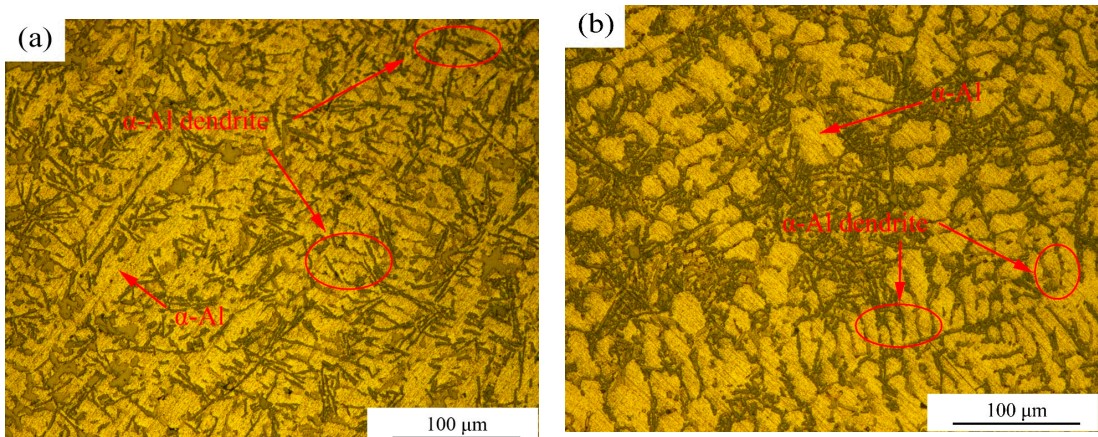

**Figure 2.** *Cont*.

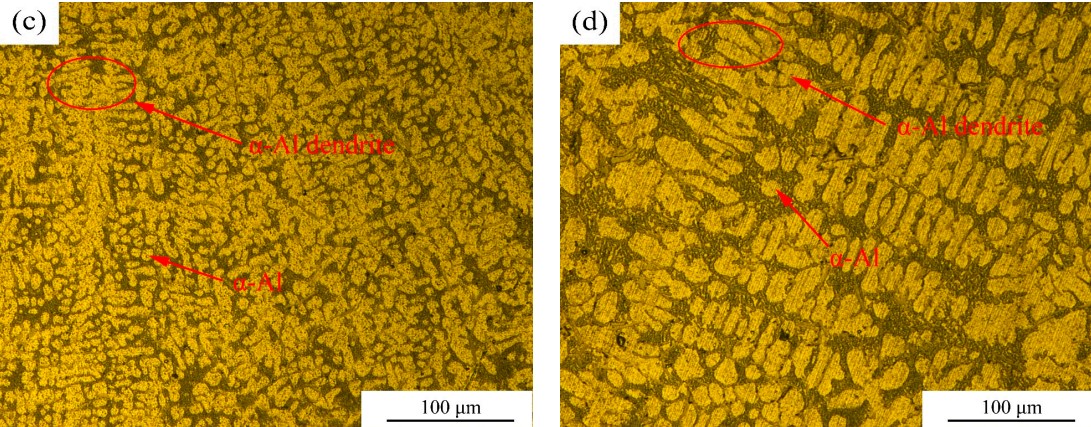

**Figure 2.** Effects of rare earth Yb addition on α-Al phases in as-cast ADC12 alloys: (**a**) unmodified; (**b**) 0.4 wt % Yb; (**c**) 0.8 wt % Yb; (**d**) 1.2 wt % Yb.

The Hume-Rothery principle suggests that only low solubility can form when the difference between the solute atomic radius and solvent atomic radius surpasses 15% [15]. The atomic radius of Al is 0.143 and that of Yb is 0.193 nm, for a difference of atomic radius of about 35%, which is far greater than 15%. As a consequence, the solubility of Yb element in the solid Al is very low, and it is difficult for Yb to enter the crystal lattice of the primary α-Al phase, instead gathering at the grain boundary and leading to constitutional supercooling. This will cause a metamorphic effect, which impedes growth and facilitates nucleation of α-Al grains [16]. There is an optimalizing spheroidized effect of the grain when the addition of Yb is 0.8 wt %. The solid-liquid interface is metastable and the primary α-Al phase tends to grow uniformly in all directions. When an excessive amount of rare earth is added, many rare earth intermetallic compounds are generated at the front of the interface, and some of the solute atoms are consumed, changing the effect of constitutional supercooling. Neighboring structures begin to overlap and encroach on each other. The grain tends to be coarser, and the SDAS increases.

### 3.2. Effects of Yb Addition on Eutectic Silicon Phases

Figure 3 presents the morphology of the eutectic Si of ADC12 with different Yb additions. As can be seen from Figure 3, Yb addition has a significant effect on the morphology of the eutectic Si. The eutectic Si morphology of the unmodified alloy is relatively coarse acicular with sharp edges, as shown in Figure 3a. The shape of eutectic silicon phases became short rod-like by 0.4 wt % Yb addition, as shown in Figure 3b. For Yb addition of 0.8 wt %, the shape of the eutectic silicon phases became small particle-like, as shown in Figure 3c. However, when Yb addition increased to 1.2 wt %, the Si phase became even coarser and the rod-like morphology reappeared, as shown in Figure 3d.

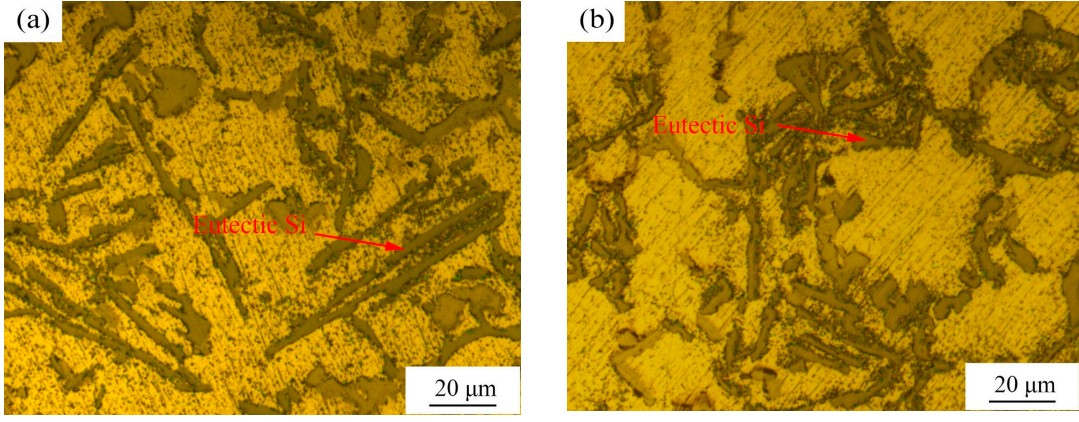

**Figure 3.** *Cont.*

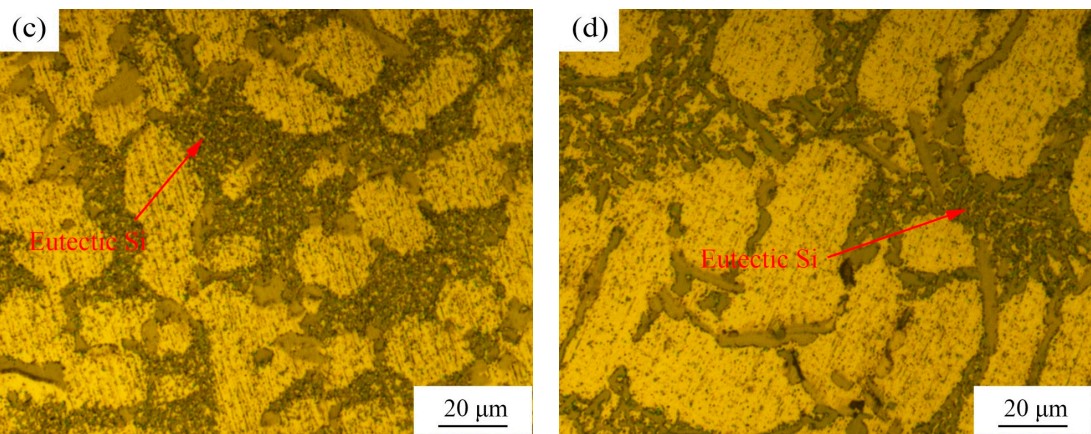

**Figure 3.** Effects of rare earth Yb addition on the eutectic Si phases in as-cast ADC12 alloys: (**a**) unmodified; (**b**) 0.4 wt % Yb; (**c**) 0.8 wt % Yb; (**d**) 1.2 wt % Yb.

Figure 4 presents the effects of Yb addition on the mean area and aspect ratio of eutectic silicon phases. The mean area and aspect ratio of eutectic Si phases are better in the Yb-modified alloys compared to the unmodified alloy. The values of the two parameters in the matrix were 36 μm² and 15.3, respectively, and the values of the two parameters in the alloy with 0.4 wt % Yb addition were 15 μm² and 7.2, respectively. When the content of Yb is 0.8 wt %, the values of the two parameters reached the minimum of 1.8 μm² and 1.1, respectively. When the content of Yb increased to 1.2 wt %, the values of the two parameters were 3.6 μm² and 2.1, respectively.

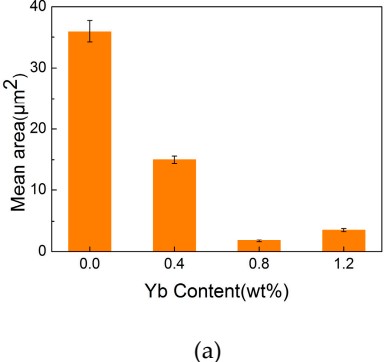

(a)

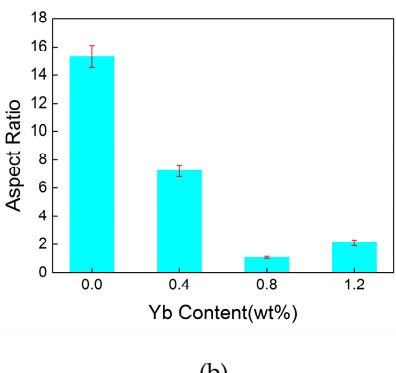

(b)

**Figure 4.** Mean area and aspect ratio of eutectic Si phases with different Yb additions: (**a**) mean area; (**b**) aspect ratio.

The metamorphic mechanism of the eutectic silicon phase by rare earth element addition is mainly the growth of silicon phases. The impurity-induced twinning (IIT) mechanism [17] is a widely accepted growth theory of silicon phases. The IIT mechanism can operate in twin plane reentrant edge (TPRE) [18] and step mechanisms. For the TPRE mechanism, chemical metamorphism is generated with addition of rare earth into the Al-Si alloys, which will poison formed reentrant angles by the twinning plane and the outer surface of the silicon crystal, thus passivating anisotropic growth of reentrant angles. According to step mode [19], IIT mode allows rare earth atoms to adsorb at the growth interface of silicon phases when rare earth is added into the Al-Si alloys, which will change stacking sequence of silicon atoms and generate some twin crystals. In this model, the growth pattern of partial positions in Si phases will change from anisotropic growth to isotropic growth. Some reports find that the concave surface of some fibrous silicon phases include fine rare earth phases, which are likely to impede the growth of Si phases, thus refining the Si phases. However, the composition of rare earth phases will consume more rare earth atoms when excessive amounts of rare earth are added, weakening the modifying effects.

### 3.3. Effects of Yb Addition on Iron-Rich Phases

Figure 5 displays the effects of Yb addition on β-Fe phases. It is clear from Figure 5a that the β-Fe phase of unmodified alloy presents a coarse lumpish morphology. However, in the Yb-containing alloys, the β-Fe phase has a finer structure, which gradually changes from bulk-shaped into short rods. As shown in Figure 5c, the finest β-Fe phase is obtained when the Yb addition is 0.8 wt %. However, the β-Fe phase becomes more coarse when the Yb addition increases to 1.2 wt %, as shown in Figure 5d; this is even more obvious in the images presented in Figure 6. The mean areas of the iron-rich phase for the Yb-free, 0.4 wt % Yb, 0.8 wt % Yb, and 1.2 wt % Yb alloys are 56, 37, 20, and 23 $\mu m^2$, respectively.

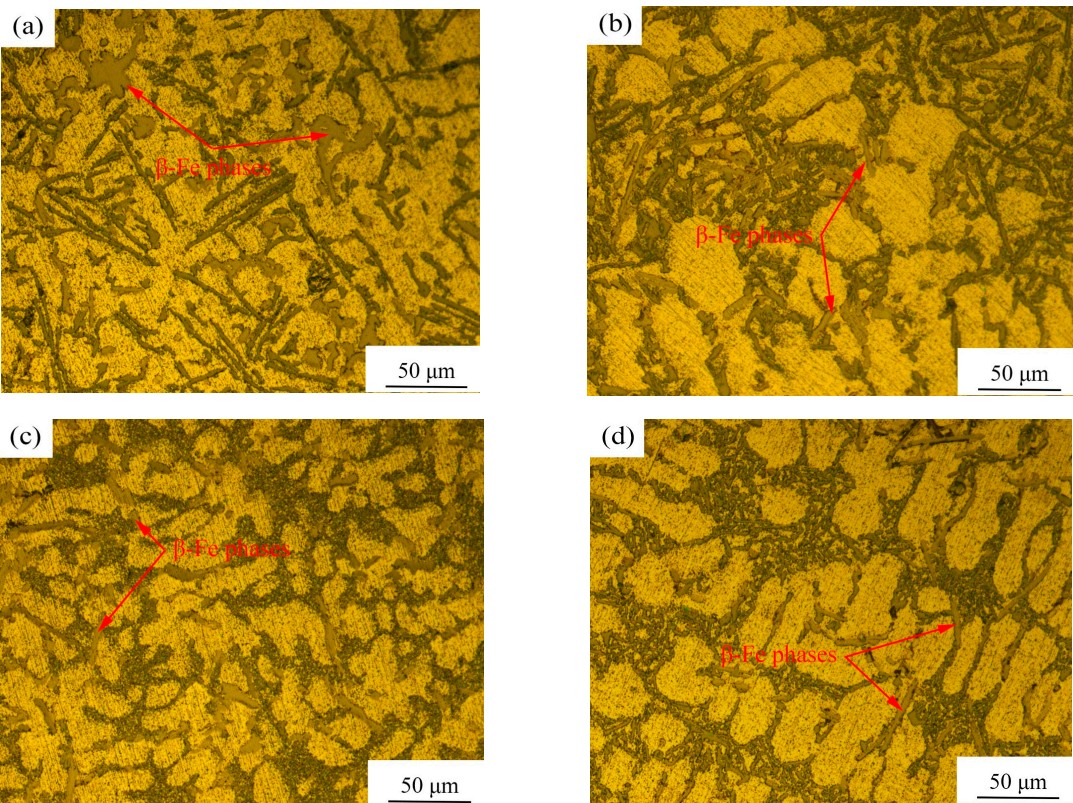

**Figure 5.** Effects of rare earth Yb addition on the iron-rich phases in as-cast ADC12 alloys: (**a**) unmodified; (**b**) 0.4 wt % Yb; (**c**) 0.8 wt % Yb; (**d**) 1.2 wt % Yb.

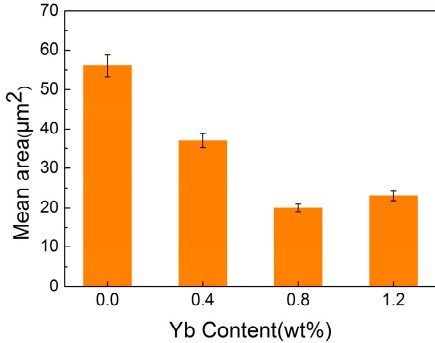

**Figure 6.** Mean area of the iron-rich phases with different Yb additions.

Figure 7 shows the SEM image and EDS results of the alloy with 0.8 wt % Yb addition. Figure 7b shows a partial amplified view of the image in Figure 7a. The point indicates the region corresponding to the EDS analysis presented in Table 2. The EDS point analysis reveals that the bright phase (marked Spectrum 1) is the rare earth phase, containing Al, Yb, and Si elements, as shown in

Figure 7c. The phase (marked Spectrum 2) mainly contains Al, Si, Fe, and Cu elements, as shown in Figure 7d. The EDS point analysis shows that the phase (marked Spectrum 3) mainly contains Al, Si, and Fe elements, as shown in Figure 7e. The phases marked Spectrum 2 and Spectrum 3 are both iron-rich phases. Some small iron-rich phases attach around the rare earth phase, suggesting that the Yb-rich phase can affect the nucleation and growth of the iron-rich phase. The stable radical of the cybotactic state [AlSiFeRE] will form when RE atoms replace Fe atoms in the radical of cybotactic state [AlSiFe]. The resulting stable radical of the cybotactic state [AlSiFeRE] can easily serve as the crystallization nuclei for the iron-rich phase [20]. As a result, some small iron-rich phases can form near the Yb-rich phase and grow along it. The rare earth phase consists of Al, Si, and Yb elements. Formation of the Yb-rich phase will consume Si atoms, but the formation of the iron-rich phase requires Si atoms, thus decreasing the overall extent of the iron-rich phase and resulting in refinement of the iron-rich phase to some extent. In addition, the solubility of Yb element in the melt is very low, implying that Yb can easily concentrate at the front of the iron-rich phase interface. This will hinder the growth of the iron-rich phase, leading to refinement. Finally, after addition of Yb, the Si phases are refined and the distance between phases is shortened. The secondary dendrite arm spacing also decreases, thus narrowing the crystal boundary [21]. Together, these factors can decrease the growth space of iron-rich phases, restricting their growth and causing refinement.

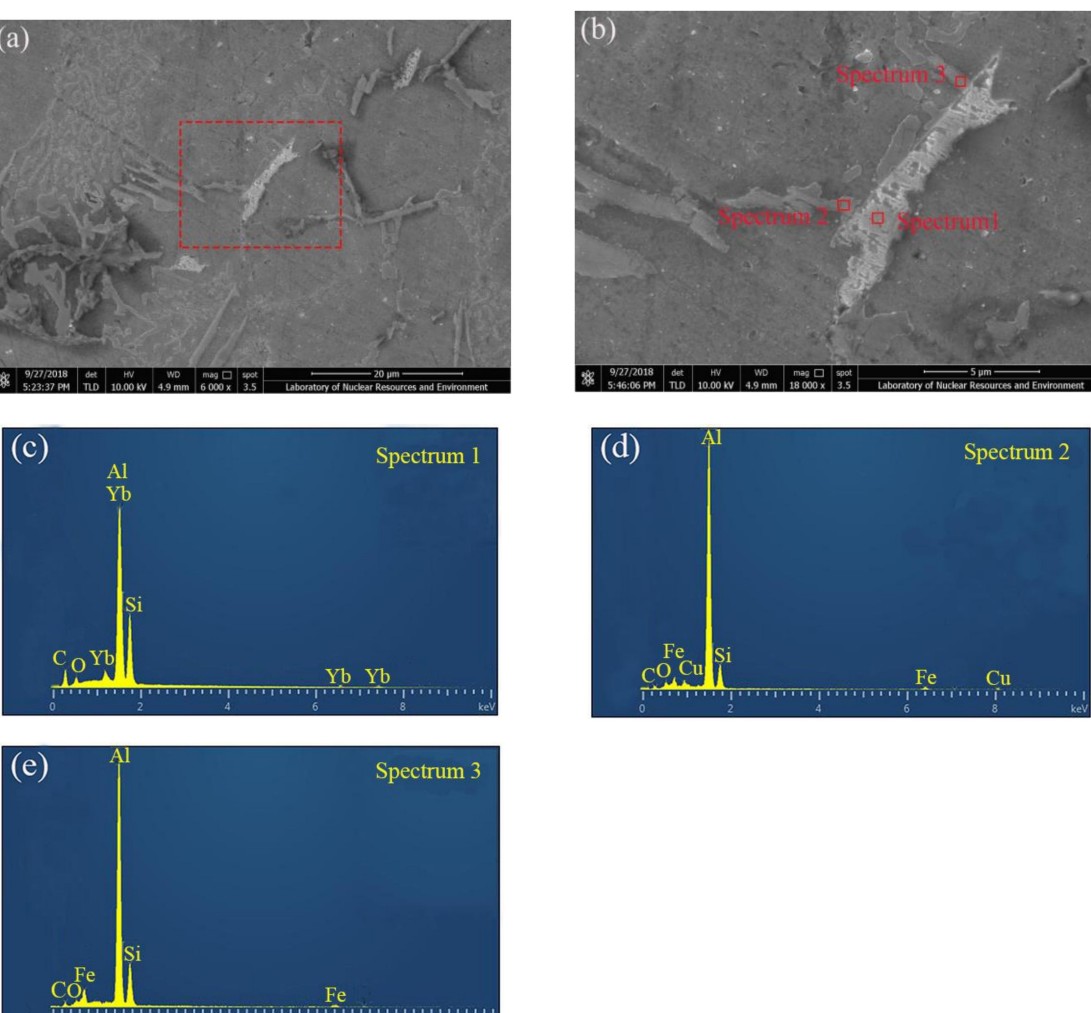

**Figure 7.** SEM images and EDS results of the alloy: (**a**) 0.8 wt % Yb; (**b**) the partial amplifying view of (**a**); (**c**) EDS of point 1; (**d**) EDS of point 2; (**e**) EDS of point 3.

**Table 2.** Chemical composition of the indicated point in Figure 7b (wt %).

| Position | C-K | O-K | Al-K | Si-K | Fe-K | Cu-K | Yb-K |
|----------|-----|-----|------|------|------|------|------|
| Spectrum 1 | 6.32 | 1.26 | 27.96 | 19.23 | - | - | 45.23 |
| Spectrum 2 | 4.53 | 2.18 | 53.44 | 17.92 | 14.34 | 7.59 | - |
| Spectrum 3 | 4.74 | 1.97 | 55.18 | 21.21 | 16.90 | - | - |

### 3.4. Mechanical Properties

Figure 8 summarizes the relationship between the microhardness and addition of Yb. The microhardness of the ADC12 alloy is only 75.9 HV but with 0.8 wt % Yb, the microhardness of ADC12 reached the maximum of 107.3 HV, a 41.4% higher value. For 1.2 wt % Yb, the microhardness of the ADC12 alloy decreased. There are two explanations for the improvement of the microhardness of ADC12 alloy with Yb addition. First, Yb is absorbed in the interface of Al-Si, strengthening the grain boundary. Second, the atomic radius of Yb is bigger than that of Al, so lattice distortion may occur, thus strengthening the matrix. When an excessive amount of Yb is added, the softening of the brittle rare earth phase will offset the partial strengthening of the solid solution, reducing the microhardness of the alloy.

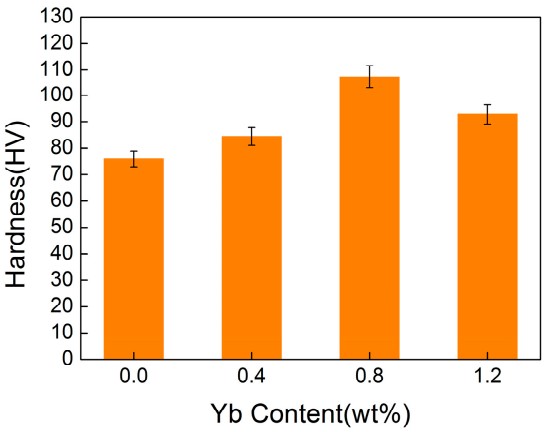

**Figure 8.** Hardness of ADC12 alloys with different Yb additions.

The tensile strengths of the as-cast alloys with different amounts of rare earth Yb were determined and are presented in Figure 9. The tensile strength and elongation of the unmodified ADC12 alloy are 172.4 MPa and 1.9%, respectively. The performance of the Yb-modified alloys are improved, with 267.9 MPa tensile strength and 4.2% elongation of the alloy with 0.8 wt % Yb addition, increases of 55.4% and 121.1%, respectively.

The mechanical behavior is controlled by the microstructure of the alloy. Combined with the analysis of the microstructure of the ADC12 alloy, the grains and the secondary phase are clearly refined for 0.8 wt %. Yb addition. Additionally, Yb addition can strengthen the grain boundary. The rare earth phase has a high melting point and hardness, and serves as an alloy-strengthening phase to improve the mechanical properties of the material. When excessive Yb is added, some phases of the alloy coarsen. Coarse phases can potentially increase the stress concentration, adversely affecting the tensile strength and elongation of the alloy.

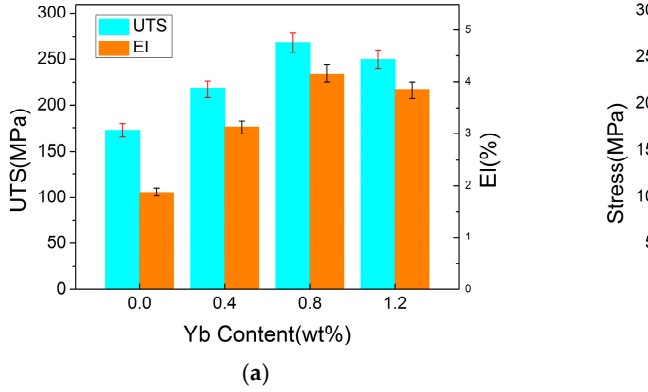
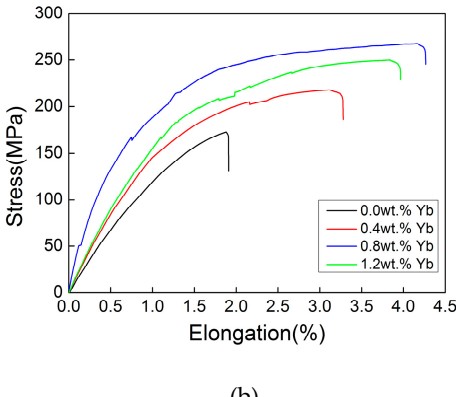

**(a)**

**(b)**

**Figure 9.** (**a**) Tensile strengths of as-cast ADC12 alloys with different amounts of rare earth Yb. (**b**) Stress-elongation curves of as-cast alloys with different amounts of rare earth Yb.

*3.5. Fractography*

Figure 10 presents the fractographs of the matrix and the alloy with 0.8 wt % Yb addition. In Figure 10a, the untreated ADC12 alloy shows typical cleavage fracture. On the surface, extensive irregular cleavage planes are visible, and there are nearly no dimples. The cleavage platforms are thick and smooth, as caused by the rupture of large eutectic silicon phases. As shown in Figure 10b, the fracture morphology of the alloy with 0.8 wt % Yb addition changes. The cleavage platforms decrease, and some dimples occurred. The dimples are small in size, regular in shape, and well-distributed, consistent with ductile fracture.

The fracture mechanism of Al-Si alloys has been attributed to three aspects [22]: the size and distribution of the silicon phases, the bonding strength between the silicon phases and the matrix, and the ease with which the Si phases crack. The coarse acicular Si phases promote stress concentration [23]. The coarse Si phases are brittle phases, which can crack and break away from the matrix. According to the Griffith Equation [24].

$$\sigma_f = \sqrt{\frac{2E\gamma}{\pi C}} \tag{2}$$

where $\sigma_f$ is maximum fracture stress, $C$ is the crack length inside the silicon phase, $\gamma$ is the interface energy, and $E$ is the Young's modulus of silicon phase. When the interior of the silicon phases cracks, the maximum fracture stress of the silicon phases can decrease. This will allow the rapid expansion of cracks throughout the silicon phases. As shown in Figure 3a, coarse needle-like eutectic Si phases are dispersed in the alloy. When the alloy is subjected to external force, the long needle-like eutectic Si phases will rotate, causing internal stress concentration around the eutectic Si phases. The maximum fracture stress of the silicon phases is reached quickly, causing breaking of the alloy.

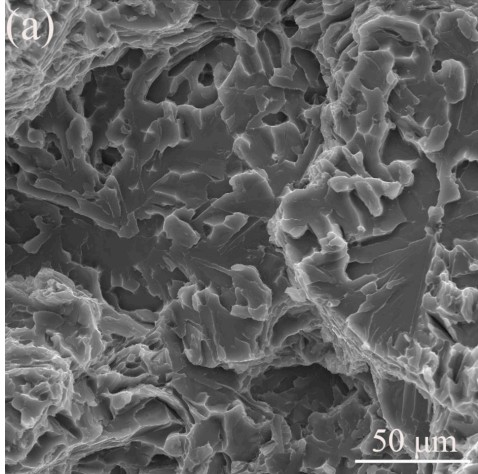
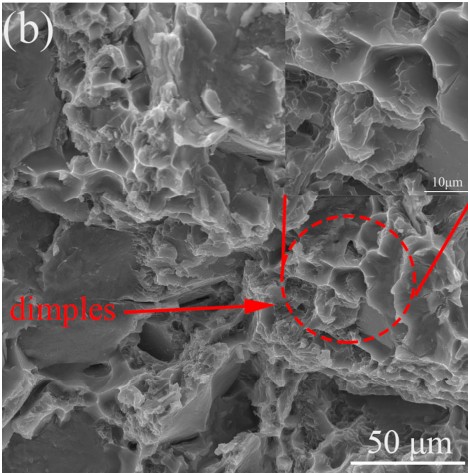

**Figure 10.** Fractographs of the tensile samples of ADC12 alloy without Yb addition and with 0.8 wt % Yb addition: (**a**) unmodified; (**b**) 0.8 wt % Yb.

Figure 11a further analyzes the fracture surfaces of the alloy with 0.8 wt % Yb addition. Figure 11b–e show the EDS results of four points as indicated in Figure 11a. Combined with the SEM and EDS analysis, the fracture of the alloy with 0.8 wt % Yb content includes spherical $\alpha$-Al phases, acicular rare earth phases, lumpish iron-rich phases, dispersive Si phases, and some stretching holes. Generally, for tiny phases, the production of cracks requires greater stress than required for massive phases. In addition, the stress for generating cracks may exceed the bonding strength between phases if the size of the secondary phase is small enough. As a result, under a great external force, coarse phases of the alloy are first to crack. However, there are few of these phases, the distribution is nonuniform, and adjacent coarse phases are far apart, so cracks are unable to rapidly extend to the whole phase. Finally, depending on a greater external force, the cracks will extend to short rod-like phases, some small granular phases, or the stretching holes. These cracks eventually contact with each other, breaking the whole sample.

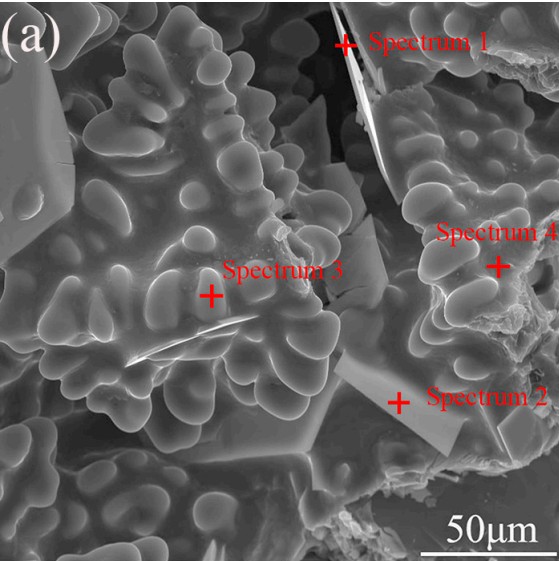

**Figure 11.** *Cont*.

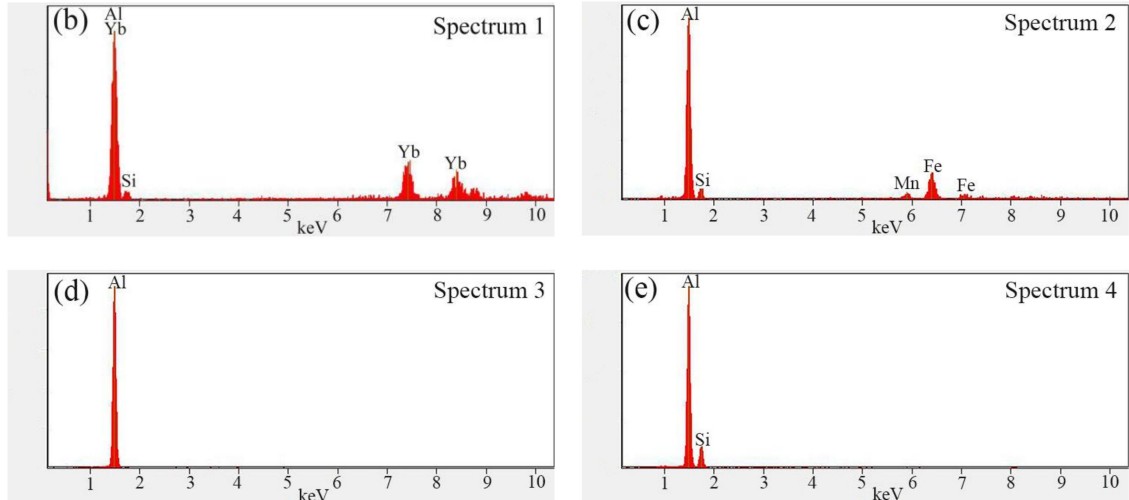

**Figure 11.** The tensile fracture analysis and EDS analysis of the alloy with 0.8 wt % Yb addition: (**a**) fractography of the alloy; (**b**) EDS of point 1; (**c**) EDS of point 2; (**d**) EDS of point 3; (**e**) EDS of point 4.

## 4. Conclusions

This study investigated the effects of Yb addition on the microstructure and mechanical properties of as-cast ADC12 alloy. The results show that:

(1) Yb is an effective modifying agent for ADC12 alloy. A Yb content of 0.8 wt % was optimal, with fully refined coarse dendritic primary $\alpha$-Al phases and decreased secondary dendrite arm spacing. The massive $\beta$-Fe phases became small rod-shaped, and the morphology of eutectic silicon phases changed from acicular into short rod-like, with some small granular.

(2) With 0.8 wt % Yb addition, the tensile strength, elongation, and hardness were 267.9 MPa, 4.2%, and 107.3 HV, respectively. These values are increased by 55.4, 121.1, and 41.4%, respectively, compared with those of the unmodified ADC12 alloy. Fractographic examinations revealed that the fracture mode is mainly ductile fracture. The mechanical properties of the alloy decrease for addition of excessive amounts of Yb. The fracture appearances are consistent with the tensile properties.

(3) The addition of Yb can generate a rare earth phase consisting of the three elements of Al, Si, and Yb, with some small iron-rich phases attached around the rare earth phase.

**Author Contributions:** J.X., H.Y., and S.Z. conceived and designed the experiments; J.X., H.Y., S.Z., and M.B. performed the experiments; J.X., H.Y., S.Z., and M.B. analyzed the data; J.X., H.Y., and S.Z. contributed reagents/materials/analysis tools; J.X., H.Y., and S.Z. wrote the paper.

**Funding:** Projects (20181BAB206026 and 20171BAB206034) supported by the Natural Science Foundation of Jiangxi Province.

**Conflicts of Interest:** The authors declare no conflict of interest.

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
