# Peer review of "Effects of Yb Addition on the Microstructure and Mechanical Properties of As-Cast ADC12 Alloy"

_metals, doi:10.3390/met9010108_

Round 1

Reviewer 1 Report

The present work presents the effects of Yb addition on microstructure and mechanical properties of as-cast ADC12 alloy. The authors have been used a sequence of characterization techniques (Optical, SEM, Hardness, Stress-Strain) to study the effect of different amount of Yb (0%, 0.4%, 0.8% and 1.2%) on the morphology of matrix, and as consequence in the behavior of mechanical properties. This theme is not new, however some of the explanation given by the authors have an attempt to clarify the mechanisms involved on the microstructure modification.

There are some remarks that need to be considered:

Major Reviewer:

 (1) Experimental Chapter. A better detailing of set-up should be introduced in the work. For instance: what are the dimensions of casted sample? Did the authors performed the degassing of liquid? Crucible dimensions and weight o ADC12 charge?

(2) According the Figure 1, it can be observed that here are differences on the modification of eutectic Silicon as well as refinement of alpha-Al, when compared the microstructure for all experimental conditions (different amount of Yb addition). However, such differences seem extremely significantly in the case of figure 1(c). The cooling rate of cast was the same for all casting?

(3) By the Figure 1(d) seems that although the grain was not refined, the modification of eutectic Silicon was effective, as can be observed in Figure 3(d) and in Figure 9(b). Can it be better explored?

(4) he authors have been quantified the SDAS (Figure 2), however according Figure 1 seems that there is presence of DAS. Can the authors clarify?

(5) The mechanism involved in the iron-rich phase refinement need to be clarify. A better sample characterization using different techniques (for instance EBSD) can help.

(6) Mechanical properties chapter. According to bibliographic data, the values of YS, and E% of the casted samples suggest being higher. Can the authors explore this issue? Why the values is so high?

Minor Reviewer:

(1) The Quality of pictures is poor.

(2) Should be introduce the name of Chapter – Conclusions.

(3) Some technical terms should be modified. For instance: Pag 2 line: 60 - into the meit in batches; Pag 3 line: 90 - microstructure of ADC12 is disordered.

(4) Small improvements in English type.

Author Response

The revisions suggested by the reviewers are very professional and detailed. We benefit much from these. It is important for us to improve the ability of research. We have revised conscientiously the paper. The changes of this manuscript have been highlighted by using the "Track Changes" function in Microsoft Word. The responses to all comments by the editorial office and the referees are provided (point by point) as below:

Reviewer 1:

Comments and Suggestions for Authors

The present work presents the effects of Yb addition on microstructure and mechanical properties of as-cast ADC12 alloy. The authors have been used a sequence of characterization techniques (Optical, SEM, Hardness, Stress-Strain) to study the effect of different amount of Yb (0%, 0.4%, 0.8% and 1.2%) on the morphology of matrix, and as consequence in the behavior of mechanical properties. This theme is not new, however some of the explanation given by the authors have an attempt to clarify the mechanisms involved on the microstructure modification.

There are some remarks that need to be considered:

Major Reviewer:

(1) Experimental Chapter. A better detailing of set-up should be introduced in the work. For instance: what are the dimensions of casted sample? Did the authors performed the degassing of liquid? Crucible dimensions and weight o ADC12 charge?

Response: Thanks for your suggestion. The ADC12 alloy was melted in the preheated graphite crucible of 60-mm diameter and 100-mm height. The aluminum alloy used for melting was about 180 g each time. The aluminum melt pool was protected by argon gas in the whole experiment. The shape and dimensions of the tensile specimen was schematically shown in Fig.1. And the related description and figure information have been added in the article now.

(2) According the Figure 1, it can be observed that here are differences on the modification of eutectic Silicon as well as refinement of alpha-Al, when compared the microstructure for all experimental conditions (different amount of Yb addition). However, such differences seem extremely significantly in the case of figure 1(c). The cooling rate of cast was the same for all casting?

Response: Thanks for your question. All the experiments were completed in the same condition. The cooling rate of cast was about 15℃/min. The Hume-Rothery principle suggests that only low solubility can form when the difference between the solute atomic radius and solvent atomic radius surpasses 15%. It is noted that the atomic radius of Al and Yb are 0.143 and 0.193nm, respectively, their difference of atomic radius is about 35%, which is far more than 15%. As a consequence, the solubility of Yb element in the solid Al is very low, Yb almost can not enter the crystal lattice of primary α-Al phase, but instead gather at the grain boundary, leading to constitutional supercooling. It will cause a metamorphic effect, which impedes the growth and facilitates nucleation of α-Al grains. There is an optimalizing spheroidization effect of the grain when the addition of Yb is 0.8wt.%.

(3) By the Figure 1(d) seems that although the grain was not refined, the modification of eutectic Silicon was effective, as can be observed in Figure 3(d) and in Figure 9(b). Can it be better explored?

Response: Thanks for your question. The optimal ratio of the atomic radius between the rare earth and the Si is about 1.646. For Yb, atomic radius ratio between Yb and Si is 1.66, the value is close to the optimal ratio, it can speculate that Yb has a fine modifying efficiency on Si phases. Besides, Metamorphic atoms can cause eutectic silicon to be produced during growth Produce more twins and exhibit a rough interface growth trend, and change the growth direction of the silicon phase, From the original sheet-like way to multi-directional growth, which affects its final shape, that is, from long needles to Granular or fibrous. However, the refinement of α-Al grains results from constitutional supercooling mainly. In general, the refinement effect of rare earth on silicon phases is more significant than that on α-Al phases.

(4) he authors have been quantified the SDAS (Figure 2), however according Figure 1 seems that there is presence of DAS. Can the authors clarify?

Response: Thanks for your suggestion. We have already marked the secondary dendrite arm in the original Figure 1. Since the proportion of dendrites is small, according to your suggestion, we will not quantify the dendrite arm spacing. As a result, we have deleted the original Figure 2.

(5) The mechanism involved in the iron-rich phase refinement need to be clarify. A better sample characterization using different techniques (for instance EBSD) can help.

Response: Thanks for your suggestion. To clarify the refinement mechanism of the iron-rich phase, we added some explanations and cited an article. EBSD is a very good sample characterization technique, and this research work is very necessary and significative. However, the contents of the analysis study are relatively large and need to have an in-depth discussion. The research group will carry on further research in the follow-up research work.

(6) Mechanical properties chapter. According to bibliographic data, the values of YS, and E% of the casted samples suggest being higher. Can the authors explore this issue? Why the values is so high?

Response: Thanks for your question. All experiments in this study were carried out under conditions of filling Ar, which ensured good castability. In addition, the properties of the alloy can make a great improvement by adding a small quantity of rare earths. The optimal ratio of the atomic radius between the rare earth and the Si is about 1.646. For Yb, atomic radius ratio between Yb and Si is 1.66, the value is close to the optimal ratio. This study was done to investigate effects of Yb addition on microstructure and mechanical properties of as-cast ADC12 alloy. The good performance results from fine microstructures. The results show that Yb is an effective modifying agent for ADC12 alloy. The optimum level of Yb content is 0.8 wt.%, and the coarse dendritic primary α-Al phases are fully refined, the secondary dendrite arm spacing is decreased, the massive β-Fe phases are turned to be small rod-shaped, the morphology of eutectic silicon phases changes from acicular into short rod-like and even small granular. The good performance results from fine microstructures. With 0.8 wt.% Yb addition, the tensile strength, elongation and hardness reach 267.9 MPa, 4.2% and 107.3 HV, respectively, increased by 55.4%, 121.1% and 41.4%, respectively, compared with those of ADC12 alloy.

Minor Reviewer:

(1) The Quality of pictures is poor.

(2) Should be introduce the name of Chapter – Conclusions.

(3) Some technical terms should be modified. For instance: Pag 2 line: 60 - into the meit in batches; Pag 3 line: 90 - microstructure of ADC12 is disordered.

(4) Small improvements in English type.

Response: Thanks for your suggestion. In order to improve the quality of the image, we have improved the resolution and color of the image. The conclusions of the article have also been modified according to your requirements. We have also modified some technical terms in this paper. There have also been some improvements in English type.

Thank you very much for your careful review for this article, and we look forward to your response. 

Reviewer 2 Report

This manuscript presents interesting and significant new results.  It should be published following attention to the following points:

The English needs review by someone fluent in the language.  In a number of places the wording makes no sense, either grammatically or because a word is used as if it had a meaning different from its actual meaning, with nonsensible results.  For example, on l. 23 "obviously refined" makes no sense (this has nothing to do with refining) and on l. 28 "basically applied with their lightweight on bodywork" is incomprehensible.  There are many more examples.

Additional minor points: SDAS (l. 15), Rie/Rsi (l. 50) and EDS (l. 157) require definition.  I cannot find the "markings" on the spectra in Fig. 7 and the markings in Fig. 11 are nearly invisible and not numbered as indicated in the caption to Fig. 12.

Author Response

Reviewer 2:

Comments and Suggestions for Authors

This manuscript presents interesting and significant new results. It should be published following attention to the following points:

The English needs review by someone fluent in the language. In a number of places the wording makes no sense, either grammatically or because a word is used as if it had a meaning different from its actual meaning, with nonsensible results. For example, on l. 23 "obviously refined" makes no sense (this has nothing to do with refining) and on l. 28 "basically applied with their lightweight on bodywork" is incomprehensible. There are many more examples.

Additional minor points: SDAS (l. 15), Rie/Rsi (l. 50) and EDS (l. 157) require definition. I cannot find the "markings" on the spectra in Fig. 7 and the markings in Fig. 11 are nearly invisible and not numbered as indicated in the caption to Fig. 12.

Response: Thanks for your suggestion. The English has been reviewed by someone fluent in the language. On 1. 23, we have changed the original sentence to another sentence: “and some small iron-rich phases attach around the rare earth phase.” On 1. 28, we have changed the original sentence to another sentence: “Al-Si alloys are extensively used in the automotive industry to reduce vehicle weight.” In the article, some more examples have been corrected, and the modified parts are marked with a red font.

On 1.15, SDAS represents the secondary dendrite arm spacing. On 1.50 Rie/Rsi represents the atomic radius between the rare earth and the Si. On 1.157, EDS represents energy diffraction spectrum. In Fig. 7, we have marked the corresponding spectrum. In Fig. 12, we have also marked the corresponding spectrum.

Thank you very much for your careful review for this article, and we look forward to your response.

Round 2

Reviewer 1 Report

The reference to Figure 11 as well as the caption should be improved.

The quality of pictures is continuing poor.

Author Response

Dear Editor and Reviewer

The revisions suggested by the reviewers are very professional and detailed. We benefit much from these. It is important for us to improve the ability of research. We have revised conscientiously the paper. The changes of this manuscript have been highlighted by using the "Track Changes" function in Microsoft Word. The responses to all comments by the editorial office and the referees are provided (point by point) as below:

Reviewer 1:

Comments and Suggestions for Authors

The reference to Figure 11 as well as the caption should be improved.

The quality of pictures is continuing poor.

Response: Thanks for your suggestion. We have merged the original Figure11 and Figure12. The caption has been improved, as shown on line 245-246. The corresponding text descriptions in the article have also been changed, as shown on line 230-233. In order to improve the quality of pictures, we have improved the resolution, the size of the font and color of pictures.

Thank you very much for your careful review for this article, and we look forward to your response.
